# Wireless Heart Sensor for Capturing Cardiac Orienting Response for Prediction of Neurodevelopmental Delay in Infants

**DOI:** 10.3390/s22239140

**Published:** 2022-11-25

**Authors:** Marcelo Aguilar-Rivera, Julie A. Kable, Lyubov Yevtushok, Yaroslav Kulikovsky, Natalya Zymak-Zakutnya, Iryna Dubchak, Diana Akhmedzhanova, Wladimir Wertelecki, Christina Chambers, Todd P. Coleman

**Affiliations:** 1Department of Bioengineering, University of California San Diego, La Jolla, CA 92093, USA; 2Departments of Psychiatry and Behavioral Science and Pediatrics, School of Medicine, Emory University, Atlanta, GA 30322, USA; 3OMNI-Net Ukraine Birth Defects Program, 33028 Rivne, Ukraine; 4Post-Graduate Extension Program, Rivne Regional Medical Diagnostic Center, Lviv National Medical University, 79010 Lviv, Ukraine; 5Khmelnytsky City Perinatal Center, 29008 Khmelnytskyi, Ukraine; 6Department of Pediatrics, School of Medicine, University of California San Diego, La Jolla, CA 92093, USA; 7Herbert Wertheim School of Public Health and Human Longevity Science, University of California San Diego, La Jolla, CA 92093, USA

**Keywords:** wireless sensor, electrocardiogram, cardiac orienting response, monitoring, neurodevelopment delay, auditory and visual stimuli, mobile application, tablet

## Abstract

Early identification of infants at risk of neurodevelopmental delay is an essential public health aim. Such a diagnosis allows early interventions for infants that maximally take advantage of the neural plasticity in the developing brain. Using standardized physiological developmental tests, such as the assessment of neurophysiological response to environmental events using cardiac orienting responses (CORs), is a promising and effective approach for early recognition of neurodevelopmental delay. Previous CORs have been collected on children using large bulky equipment that would not be feasible for widespread screening in routine clinical visits. We developed a portable wireless electrocardiogram (ECG) system along with a custom application for IOS tablets that, in tandem, can extract CORs with sufficient physiologic and timing accuracy to reflect the well-characterized ECG response to both auditory and visual stimuli. The sensor described here serves as an initial step in determining the extent to which COR tools are cost-effective for the early screening of children to determine who is at risk of developing neurocognitive deficits and may benefit from early interventions. We demonstrated that our approach, based on a wireless heartbeat sensor system and a custom mobile application for stimulus display and data recording, is sufficient to capture CORs from infants. The COR monitoring approach described here with mobile technology is an example of a desired standardized physiologic assessment that is a cost-and-time efficient, scalable method for early recognition of neurodevelopmental delay.

## 1. Introduction

Two to five percent of first-graders in the U.S. have evidence of Fetal Alcohol Spectrum Disorders (FASD) [1]. Unfortunately, many of these children are misdiagnosed, or their diagnosis is wholly missed, especially in infancy and throughout early childhood [2]. Earlier and more accurate identification of children with prenatal alcohol exposure (PAE) can lead to interventions that engage neuroplasticity associated with early brain maturation, thus affording these children the best chance of improving outcomes. However, early identification of individuals who are neurodevelopmentally impaired as a function of PAE is challenging due in part to limitations in standardized tests of neurodevelopmental functioning, especially for infants and toddlers. Existing standardized measures require substantial resources in terms of cost because of the training and expertise of clinical personnel. Research studies involving physiologic measures are also cost-prohibitive and time-prohibitive because of the need for expensive, clunky hardware, both for the collection of the heart rate data and stimulus presentation, with each of these systems requiring expert knowledge in coding specific software.

Specialized infant assessment protocols such as the cardiac-oriented response (COR) are sensitive to the impact of PAE at the early age of 6 months [3] and in toddlers [4]. The COR is characterized by heart rate (H.R.) deceleration in the presence of novel stimuli presentation [5,6], which serves as a proxy for prefrontal cortex activity reflecting learning about environmental events [7]. Given that the COR only requires the acquisition of heartbeats and acoustic/visual stimulus delivery, it can, in principle, be obtained inexpensively with limited examiner expertise.

The feasibility of using a lightweight, single-channel device as an alternative to a larger ECG has been established by another group that validated a single-channel ECG sensor against a 12-channel ECG in an exercise paradigm [8]. The R to R interval measurements generated by the single channel device were highly correlated with the larger ECG, demonstrating the device’s accuracy in detecting heartbeats. The data was also conveyed to an external app for real-time viewing.

One natural step in scaling screening tools for the early diagnosis of FASD is to develop minimally obtrusive wireless technology for sensing heartbeats and transmitting heartbeat times synchronous with the timing of acoustic and visual stimuli.

Here, we present a sensible next step in addressing the unmet needs in FASD diagnosis and quantifying neurodevelopmental delay. Specifically, we offer an inexpensive and unobtrusive single lead wireless ECG sensor in tandem with a mobile tablet application containing software to automate assessments, allowing for a description of the child’s neurophysiological response and thus enabling the comparison of the children’s performance to that of others.

The system contains a low-noise ECG acquisition system that extracts heartbeat times from the ECG and wirelessly transmits heartbeat times via Bluetooth low energy (BLE) to an iOS tablet. The tablet contains software for providing acoustic and visual stimuli, acquisition of ECG heartbeat times wirelessly transmitted from the wearable device, and time-synchronizing stimuli display with the ECG heartbeat times. In addition, a custom mobile application is developed to plot the heart rate in beats per minute (BPM) concerning time since the stimuli, providing feedback to the operators immediately after the COR test is completed. The heartbeat times, relative to stimuli, are saved in a standardized file format, allowing for remote transmission for data aggregation or for any team member to view. We have tested our heartbeat sensor within the context of a COR habituation/dishabituation paradigm involving visual and auditive stimulation on a sample of children enrolled in studies conducted in Ukraine and Atlanta.

## 2. Materials and Methods

We collected data from an ongoing longitudinal cohort study in Ukraine and a separate cohort in Atlanta. We used a state-of-the-art technology biosensor for extracting ECG, detecting heartbeats, and wirelessly transmitting to a tablet. In tandem, we developed a mobile tablet application that synchronizes with the heartbeat sensor, plays auditory and visual stimuli, presents cardiac-orienting response information to the clinical user, and stores results in a standardized format.

**COR Sensor.** We used a MAXREFDES100# health sensor platform, here referred to as the data acquisition board. Its compact internal MAX30003 biopotential integrated circuit (I.C.) is a clinical-grade single-channel integrated biopotential (ECG) chip with a high-resolution data converter, low noise amplifier, and real-time extraction of R to R intervals that allowed us to calculate the BPM. These data were sent through BLE to our custom mobile application via the wireless capabilities (BLE radio chip EM9301) of this acquisition board. The MAX30003 records ECG signals and extracts the R to R intervals in real-time. These data are sent via a serial peripheral interface (SPI) to the MAX32620 chip, an ARM Cortex microcontroller unit that is part of the acquisition board. This MAX32620 chip routes the data through an SPI to the EM9301 BLE radio chip that allows their wireless transmission to the mobile application.

We interfaced the acquisition board with two wired female snap connectors, where the main ECG lead was placed on the left chest while the ground was placed on the left lateral side of the thorax. In this way, silver-silver chloride disposable electrodes were easily attached and detached from the acquisition board. The end-to-end distance between the two wired female snap connectors was 10 cm. The acquisition board was operated on a coin cell battery CR2032 after installing a battery holder for such purpose on the bottom side of the board. The wired acquisition board with the battery holder was placed in a custom 3D printed (Creality UV curing 405 nm standard photopolymer resin) plastic case for protection, while only the two female snap connectors protruded from the plastic enclosure.

**Firmware on the Mobile Acquisition Device.** We modified the source code provided by Maxim Integrated to customize firmware enabling BLE communication between the acquisition board and our custom mobile application. The ECG signal was digitized at 128 Hz, while the firmware allowed a maximum sampling rate of 4 Hz for the heartbeat time interval (R to R) represented in 14 bits, while the timing resolution of the R to R interval was approximately 8 ms. The MAX30003 IC contains a built-in hardware algorithm to detect R to R intervals based on the modified version of a previously published adaptative algorithm for real-time QRS complexes detection [9]. Pan and Tompkins described that their algorithm reliably recognizes QRS complexes based upon digital analyses of the slope, amplitude, and width with a high accuracy.

**Cardiac Orienting Response Application on the Tablet.** Our custom application for iOS devices allows us to set different parameters associated with stimuli display, such as randomizing the stimulus modality order (visual vs. auditory) and the specific stimuli order used for habituation stimulus vs. the dishabituation within a modality. The same applies to the auditory stimuli, in which sound stimuli change during habituation and dishabituation. In the application, the user can also choose to record only BPM data and/or the raw electrocardiogram (ECG) waveforms, although ECG data recording has not been implemented at the firmware level yet.

The visual and auditory stimuli display followed the scheme depicted in Figure 1A. The habitation stimulus is presented ten times, while the dishabituation stimulus is shown only five times. We ran our mobile app on 10-inch iOS tablets (iPad air 3rd Generation, Model # MUUK2L4A, and software version 15.6.1, Apple Inc., Cupertino, CA, USA) and collected timing data regarding the elapsed R-to-R time. From there, we computed the BPM in response to habituation and dishabituation stimuli displayed as follows. For each interval k of duration 1 s, we captured the R to R intervals (in units of milliseconds) and calculated the mean R to R interval in that duration, which we term R2R [k]. Then we have that the heart rate in beats per minute for interval k, given by BPM [k], satisfies:BPM [k] = 60,000/(R2R [k])(1)

The R-to-R data described in Equation (1) was displayed in the application as the number of BPM before starting with the stimuli presentation. In this way, the practitioner can determine if the acquisition board is properly extracting the infant’s heartbeats. Additionally, some basic processing, such as averaging the deceleration of the first three trials for the habituation and dishabituation conditions (Figure 1A), was applied and displayed in the same application to provide feedback to the practitioner regarding the child’s encoding of the stimuli. Our custom mobile application also allows the R-to-R timing data to be saved as a standardized comma-separated values (CSV) file for additional analysis that can be shared through the internet (i.e., by email or via another application). Further analyzes were then conducted on the data to (i) quantify the quality of the beat times data and (ii) quantify the heartbeat deceleration in response to the novel stimuli (Figure 1B) as described later in the manuscript.

**Infant car seat and tablet-based stimuli display setup**. An infant car seat was interfaced with a tablet arm mount (Tabletop Arm Mount—CTA) that allows three degrees of freedom in the X, Y, and Z axis regarding the position of the tablet display. This setup provides secure and comfortable seating for the infant in front of a 10 inch iOS tablet, where visual and auditory stimuli were presented while we recorded the heartbeat rate using our custom COR sensor and mobile application.

The acrylic base was 15″ wide, 23″ deep, and 3″ high, and its top surface was covered with a rubber mat to prevent the infant car chair from sliding. To improve safety, the car seat can be secured to the acrylic base with the help of rubber bungee cords with J-hooks. It is possible to set two sizes of infant car seats on the acrylic base as follows: (a) a seat for infants of 4–22 pounds (1.8–10 kg) and a maximum height of 29 inches (74 cm); (b) a larger chair for infants of 22–40 pounds (10.1–18 kg) and a maximum height between 29–43 inches (73.6–110 cm).

Since the arm mount that held the tablet was flexible, it was possible to easily adjust the distance from the infants. The 10″ tablet was placed in front of the infants at a distance of 30 cm (12 inches) from their heads, at the level of their eyes, while the infant was seated on the car seat.

**Stimuli Display Paradigm.** A 30-s baseline period of heart rate was collected prior to the initial stimulus presentation. The standard auditory stimuli consisted of alternating 400-Hz and 1000-Hz pure tones presented contiguously for 2 s each, with three repetitions, so each auditory trial lasted 12 s. The novel auditory stimulus consisted of alternating 700-Hz and 1000-Hz pure tones with three repetitions per trial. The standard visual stimuli consisted of the chromatic Caucasian faces of a baby, while the novel visual stimulus was that of a woman. The standard and novel stimuli are exchangeable inside each stimulus modality as well as the modality order (visual vs. auditory). The standard stimulus for habituation was presented for a total of 12 s, followed by an interstimulus interval of 12 s until ten repetitions were completed. The novel stimulus for dishabituation was then presented under similar timing conditions (12 s with a 12-s interstimulus break) for five trials. The total duration of the habituation and dishabituation procedure was 12.5 min since the six minutes for each stimulus modality plus the 30-s baseline.

Only the first three trials of the habituation and dishabituation session were used to analyze the data, given a significant COR attenuation has been reported starting at the fourth trial of exposure for the habituation and dishabituation stimulus [3].

## 3. Results

Here we present a low-cost, user-friendly system to capture the magnitude of the heart rate deceleration during CORs from different populations of interest, such as infants at risk of neurodevelopment delay. We anticipate that this methodology will be helpful in identifying those with neurodevelopmental delay associated with prenatal alcohol exposure, but more importantly, we recognize this methodology may also identify potential neurodevelopmental delay resulting from other etiologies that also impact prefrontal cortical responses related to encoding environmental events. The main goal of using this system is to improve early identification of those with neurodevelopment delay so that habilitative care can be initiated as early as possible [10]. Children with neurodevelopmental compromise have been shown to benefit from early interventions, such as cognitive stimulation and speech, occupational therapies, and physical therapies, that can prevent a cascade of secondary adaptive skill deficits [11].

Figure 1A depicts the temporal sequence of visual stimuli presentation for a total of fifteen trials. The duration of each visual stimulus presentation was 12 s. A similar total time is used for the presentation of the fifteen auditory trials. Given the flexibility of our system, it is possible to randomize which stimulus modality is presented first (visual vs. auditive) and the specific stimuli arranged for use as the habituation stimulus vs. the dishabituation stimulus within each modality.

At the COR sensor level, the single lead ECG is a tiny device that runs on a disposable coin cell battery CR2032 (Figure 2A) that can be easily replaced as needed, as well as the silver-silver chloride electrodes. Given power consumption optimization was not an aim at this point, the whole circuit from the amplifier to the BLE radio takes a considerable amount of current while capturing ECG signal and, most importantly, broadcasting data wirelessly. We suggest recording a maximum of three experiments per each new and fully charged CR2032 battery.

As mentioned below, the mobile app (Figure 3) is very versatile regarding stimuli randomization. The user can specify the stimulus modality order (visual vs. auditory first) and, within each modality which of two different stimuli serves as the habituation or dishabituation stimuli. Additionally, the mobile displays some primary graphical statistic at the end of the habituation-dishabituation protocol. The app also allows sharing files using the Apple software “Music” (formerly “iTunes” for previous versions to macOS Catalina) when the tablet is connected to a computer through a USB cable in the case of iOS. By using macOS Monterey, it is possible to extract the data saved by the COR app as a CSV file directly by selecting the tab “data” after clicking on the tablet mounted as an external drive on the iOS system when it is connected to the computer through a USB port.

Regarding the remaining components of the system described in this article, an infant car seat was interfaced with a flexible tablet arm mount that allows several degrees of freedom to properly adjust the distance between the tablet and the infant tested (Figure 4). We suggest a separation of 12 inches from the infant to the screen tablet, as was previously used in our research [3,6].

Figure 5 depicts different tests of the COR sensor when biopotentials and synthetic signals with other parameters were used. In short, we demonstrated that our system is able to capture COR as those ECG commercial systems used in previous studies [3,6]. In those studies, it has been shown a drop of the heart rate in a range of 6–12 BPM occurs in the peak-trough phase of the COR, which matches the six BPM that is quantified here and depicted in Figure 5C, in response to a change in the frequency of the synthetic signal feed to the COR sensor. Most importantly, we applied to the sensor ramps of artificial ECG signals on steps of 0.1 Hz, equivalent to 6 BPM per step, with a duration of 12 s and 1 mV of amplitude. In this way, we evaluated the response of the COR sensor in terms of variability from 1.4 Hz to 4 Hz, including the median heart rate (~140 BPM) characteristic of infants between 3 to 6 months old [12]. It is possible to see that the response of the COR sensor is stable at different frequencies having a coefficient of variation (CV) < 1% when the input signal is <3 Hz, making this system suitable for the study of R to R on infants at any age. It is possible to see the level of noise, here reflected on the CV, increased when the frequency of the input signal was over 3 Hz, which is equivalent to a heart rate over 180 BPM. Additionally, we tested the COR sensors using similar ramps of frequencies but at different amplitudes. We tested from 0.5 mV to 1.5 mV, given it is known that the R wave amplitude is ~1.5 mv on infants 1 to 3 months old [13]. The COR sensor mainly responds linearly to synthetic ECG signals from 1.4 to 3 Hz, equivalent to 84 to 180 BPM, respectively, matching the electric parameters of biological ECG signals. We demonstrated that our COR sensor can capture variations on the synthetic input signal of 0.1 Hz (grey curve in Figure 5C) through time, making this sensor suitable for detecting differences in the heartbeat rhythm and between different populations of interest, such as healthy vs. neurodevelopment delayed subjects.

In this regard, we show the COR sensor described in this article is able to capture the biological R-to-R variations, expressed here as BPM, during a protocol of stimulus habituation-dishabituation on infants. In fact, Figure 5E,F show the heartbeat deacceleration in response to a habituation-dishabituation protocol for visual as well as auditory stimuli. This result clearly indicates that our system can capture COR and its modulation related to repeated stimulus presentation. The shape of the drop on the R to R in response to the visual and the auditory stimuli, as well as its timing, matched well to that described previously by our team [2,4,14] when the same protocol and readout were used, but different commercial ECG system.

## 4. Discussion

We have presented in this work a device and mobile application that, in tandem, enable physiologic recording data wirelessly, presenting auditory and visual stimuli on an iOS tablet, and placement of a tablet at the desired distance from an infant using a car seat and minor modifications with 3D printed plastic parts that altogether expand the contexts in which infants can be monitored with the COR paradigm. The preliminary data suggests the method used is a promising achievement for the implementation of a standardized physiological developmental test based on neurophysiological modulation of cardiac activity in response to environmental events.

The results presented in this article are backed up indirectly by recent findings published regarding a single-channel ECG system for R to R interval detection and heart rate variability (HVR) analysis based on hardware built on the same clinical grade biopotential chip used in our system [8]. Even though in such work, the analytic steps are not exactly the same as those followed here, the results are similar regarding the accuracy of a single lead system to detect R to R intervals. The authors reported a high degree of correlation (Pearson’s r between 0.98 to 1.0) for the mean R to R between the single lead system and a conventional 12-channel ECG [8]. Additionally, a recent systematic review and meta-analysis demonstrate that HRV measurements acquired from portable devices show a small amount of absolute error compared to the “gold standard” ECG [15]. The authors concluded that this small error in accuracy is acceptable when considering the improved usability and user application compliance of HRV measurements and analysis acquired through low-cost portable devices for remote application in health and disease. Additionally, the authors of the original real-time algorithm for QRS detection on which the built-in algorithm in the Max30003 is based, reported a 99.3% of correct detection of QRS complexes [9].

COR is a time-effective diagnostical approach for early recognition of neurodevelopmental delay. The early identification of infants at risk of neurodevelopmental delay is of public health interest, as it allows for early interventions that take advantage of the neural plasticity in the brain under development. Our system is not only a time effective but also a promising cost-effective tool for early recognition of neurodevelopmental delay. In this regard, the production by volume (i.e., thousands of units) of a similar prototype could cost less than twenty dollars, considering the bill of materials, fabrication of the boards, and assembly of the circuit itself. One advantage of such a system is that having snap connectors mounted on it makes it possible to attach and detach silver-silver chloride disposable electrodes, making this system reusable. Another important detail is that having a battery holder makes it possible to replace the coin cell battery as much as needed. Finally, having a USB port mounted on the board provides an easy method to update firmware by connecting the sensor board to a computer, as well as extracting the captured COR data. This type of inexpensive wearable configuration increases the opportunity for this technology to be quickly adopted in low-resource communities worldwide.

The COR’s value and validity as a predictor of prenatal alcohol exposure have been demonstrated across a range of studies. The magnitude of the deceleration in heart rate relative to a baseline response during a COR to visual stimuli is associated with drinking patterns around the time of conception [4]. Similarly, the magnitude of the heart rate deceleration during CORs collected at six and 12 months have been found to predict a neurodevelopmental delay in preschoolers [6], and CORs have demonstrated predictive ability that competes with commonly used and more resource intensive measures [14]. CORs may have utility in corroboration with other assessment tools and as standalone measures. For the purposes of producing accurate and effective technology, CORs can be reliably extracted in real-time to differentiate between subjects with and without PAE. Given the extensive literature that has been established on the COR being used as a predictor of neurodevelopmental delay in children with Fetal Alcohol Spectrum Disorders, this framework has great potential to address key unmet needs in monitoring more children so that they can obtain early intervention services as soon as a possibility if the neurodevelopmental delay has been identified. Recently developed machine learning methods that operate on the COR data and have shown the ability to predict downstream neurodevelopmental delay shows excellent promise to be used in conjunction with these mobile technologies [16]. These scalable methods being tested have global public health relevance as they are feasible to implement in limited-resource settings.

Furthermore, this measure can be obtained in infancy and with minimal resources or particular expertise required for implementation, lending itself to broad-based scalability if appropriately packaged for clinical practice.

## Figures and Tables

**Figure 1 sensors-22-09140-f001:**
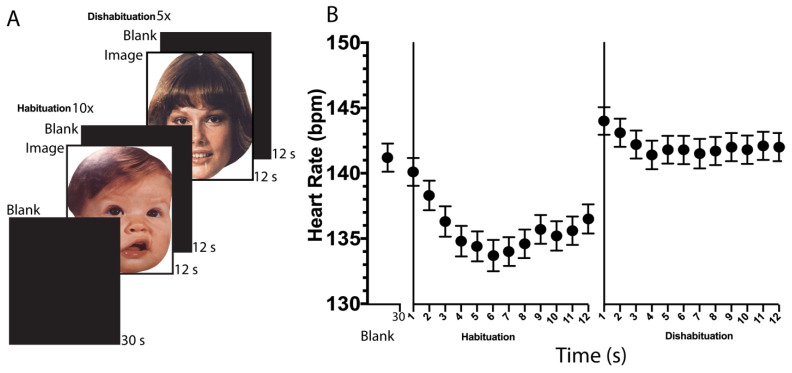
The cardiac-oriented response is triggered by stimulus display. (**A**) The visual stimulus category was composed of the presentation of a blank screen followed by the picture of a baby or woman’s face and then a blank. In this pictorial example of the habituation session, a baby face picture was presented ten times. Then, during the dishabituation session, the other face picture was used; in this example, a woman’s face followed by a blank was presented five times. Repetitions and duration that each visual stimulus displayed are depicted at the top-left and bottom-right corners of the pictures, respectively. (**B**) Previous unpublished data representing a canonical averaged heartbeats deceleration during the first three images presented for the habituation (baby face picture) and dishabituation (woman face) visual stimuli. Data were collected with a commercial setup, different from that described here. Dots and bars represent mean and standard error, respectively, calculated from 162 subjects with no prenatal alcohol exposure.

**Figure 2 sensors-22-09140-f002:**
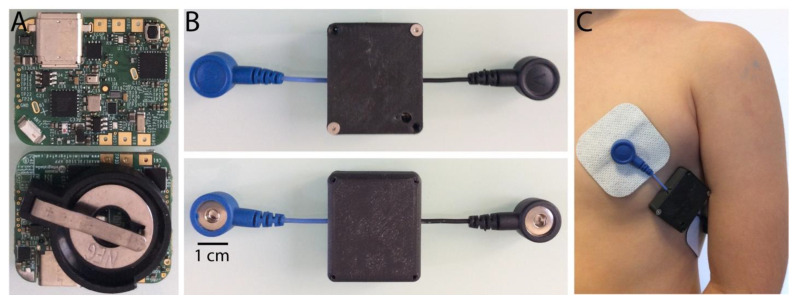
COR sensor. (**A**) Top view of the board showing the ECG components and BLE set chip for wireless communication. The bottom view shows the battery holder for a CR2030 coin cell. (**B**) The end-to-end length includes ground (black), main lead (blue), and the 3D-printed black plastic encase that protects the primary circuit. (**C**) An example of leads positioning on a 4-year-old child.

**Figure 3 sensors-22-09140-f003:**
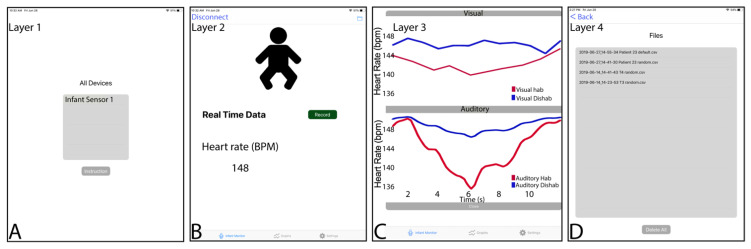
COR mobile app running on a 10” iOS tablet. (**A**) The primary layer shows a COR sensor listed during BLE advertising, just before the connection between the sensor and the app is established. When the user clicks on the COR sensor name listed under “All Devices,” such a connection between both ends is established. (**B**) Since the connection between the sensor and the app, the user can see the heart rate information displayed on the left under “Heart rate (BPM).” (**C**) When the experiment is completed, the user gets feedback regarding deceleration as the averaged BPM for the first three trials for habituation and dishabituation at each stimuli category. (**D**) This final layer shows the experiment stored in the iOS device and provides the option to select and share a selected experiment by email.

**Figure 4 sensors-22-09140-f004:**
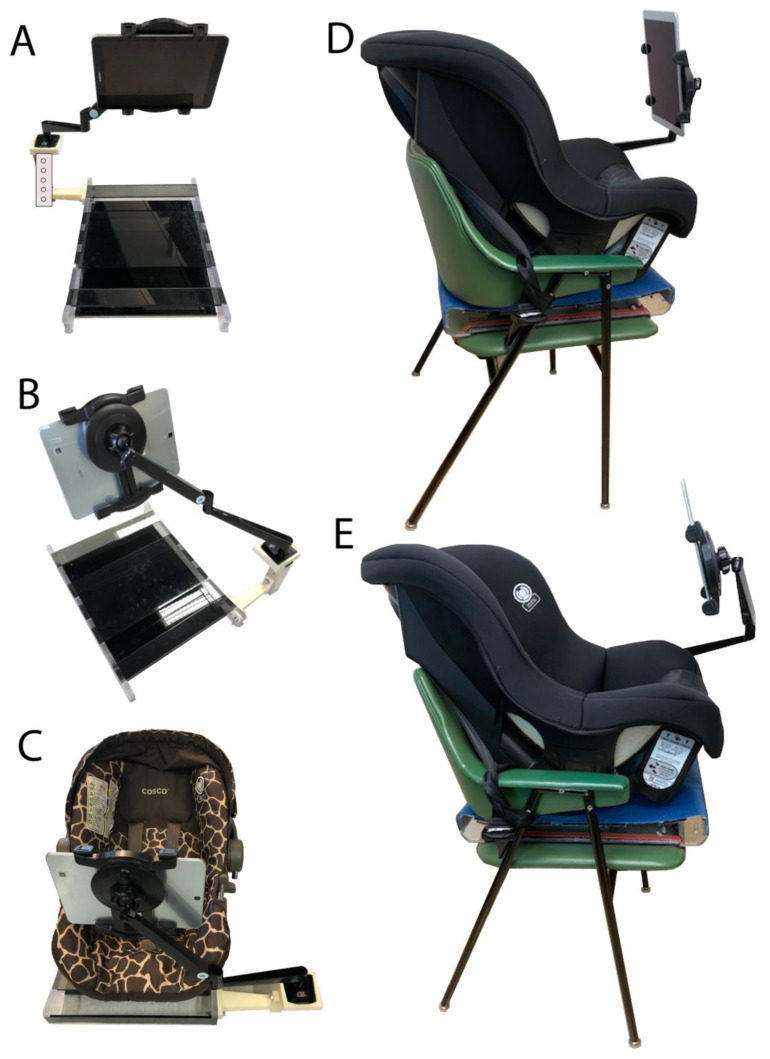
Car chair support and tablet holder system. (**A**,**B**): Rear and frontal view of the mechanical frame attached to the acrylic base that allows to hold and accommodate a tablet in front of the infant seat on the car chair. (**C**): Car seat for infants of 4–22 pounds. (**D**,**E**): Lateral views of the same system support a larger car seat for infants of 22–40 pounds and a tablet while the system is placed on the top of a high chair. Note: the chairs used on this figure are examples of how we took advantage of combining such chairs and the COR system. In the text we provided suggestions on how to attach both components, however the use of chairs with the COR system is under your own risk.

**Figure 5 sensors-22-09140-f005:**
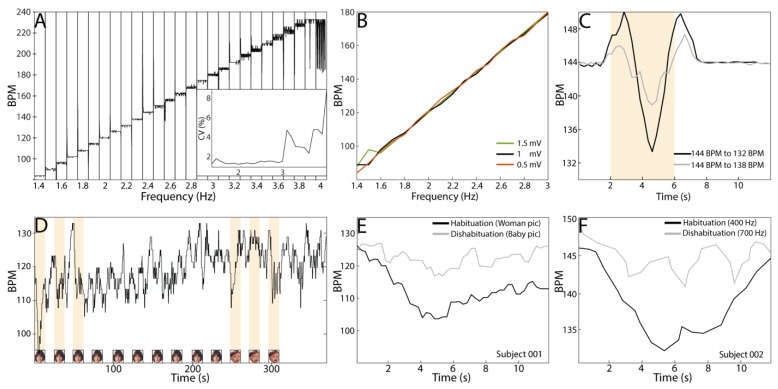
Response of COR sensor to synthetic and heart biopotentials. (**A**): The average curve of two ramps of artificial ECG signals on steps of 0.1 Hz is equivalent to 6 BPM every 12 s. It is possible to see the response of the COR sensor is stable at different frequencies having a coefficient of variation (CV in the insert) < 1% when the input signal is < 3 Hz. (**B**): The COR sensor mostly responds linearly to synthetic ECG signals at different frequencies (Hz) and amplitudes (mV) that match the parameters of biological ECG signals. (**C**): The COR sensor can capture variations of 4 s (yellow column) or longer on the synthetic input signal of 0.1 Hz (grey curve) through time, making this sensor suitable to detect differences in the heartbeat rhythm between populations (i.e., healthy vs. neurodevelopment delayed subjects). (**D**): Raw R-to-R time expressed as BPM during a whole session of visual stimulation. (**E**): COR triggered by the visual stimulation depicted in D where the woman pictures were used as the habituation stimulus, while the baby pictures were used as the dishabituation stimulus. The black curve represents the averaged cardiac deceleration in response to the first three visual habituation images (yellow columns) of an infant with no prenatal alcohol exposure from an Atlanta cohort. (**F**): The black trace represents the averaged cardiac deceleration in response to the first three 400 Hz habituation tones of a subject with no prenatal alcohol exposure during gestation from the Ukraine cohort. The gray curves are the averaged COR to the first three dishabituation pictures (baby face) and tones (700 Hz) for each subject and stimulus modality, respectively.

## Data Availability

Experimental data are available upon request.

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
