# Peer review of "Wireless Heart Sensor for Capturing Cardiac Orienting Response for Prediction of Neurodevelopmental Delay in Infants"

_sensors, 2022, doi:10.3390/s22239140_

Round 1

Reviewer 1 Report

This looks like an interesting system and all aspects of the system and experimental setup are clearly described. My main issues are that the system requirements for this application aren't clear and that the results presented are insufficient to say whether it works.

recommendations to improve the paper would be:

Introduction

1.1 It would be useful to quantify the size of the effect you are looking to spot.

1.2 The movesense system sampled ECG at 512hz and the data was analysed, substantially processed offline and the heartrate was extracted. So while both this system and the movesense system are single channel systems using the same front end, the movesense paper (reference 8) doesn't support the approach here of using the built in RR detection.

Method

2.1 Some discussion of latency is needed due to on chip processing, buffering and the wireless link. Although this should present as a systemic error for both control subjects and subjects with PAE

2.2 Some analysis and validation of R-R errors is needed, as the on-chip algorithm has a resolution of about 8ms (an error of 3bpm at 150bpm) and potentially incurs signficant errors due to the method it uses to approximate the QRS complex.

Results

3.1 I would need to see R-R data gathered from this system compared to some form of gold standard system.

3.2 I would also very much like to see more of the habitation/dishabitation data from Figure 5, e.g. habitation and dishabitation traces from multiple subjects. Having n=1 isn't useful, and multiple trials with the same subject wouldn't be much better.

Reviewer 2 Report

Review of “Wireless Heart Sensor for Capturing Cardiac Orienting Response for Prediction of Neurodevelopmental Delay in Infants” for Sensors 1832771.

The authors describe a novel wireless heart rate monitor and a novel software application for tablets, whose synchronized operation can run infant cardiac orienting response (COR) experiments for the purpose of early detection of impaired neurological function. The device is portable and lightweight, based on commercial chipsets, and is reusable by interchangeable interface electrodes, battery, and upgradeable firmware. The software application communicates with the wireless device and can run synchronized visual and auditory stimulation, together with live data collection of the cardiac pulse interval that is used to calculate infant CORs. They use this system to record auditory-induced COR from one infant. This work addresses an important need because it is argued that early detection of abnormal CORs would be beneficial to apply useful therapeutic interventions.

This manuscript introduces well-engineered device and software that seem appropriate and promising for the proposed application. However, some flaws prevent its publication in its current state.

1.       In the project motivation presented in introduction and abstract, it is unclear what will be gained by identifying infants with abnormal CORs. In particular, it is not specified what effective intervention could be applied, or what outcome improvement could be possible. The authors could elaborate on this and add references to support the “unmet need” that is mentioned. At the very least, a reference should be added to support the statement “can lead to interventions that engage neuroplasticity”, and “engage neuroplasticity” should be rephrased or explained to indicate a specific outcome improvement.

2.       The device is well described, but validation is not enough to assess whether the device produces recordings that will be useful or not. While recordings from more than one subject would be ideal to validate its robustness, the existing validation data is also unclear. Figure 5 presents the COR response of one subject. Authors should add standard error to the plots to understand the trial-to-trial variability. Authors should add interpretation to this data: Are the shapes of the responses consistent with those presented in Figure 1B? Are the differences between habituation and dishabituation significant? Should the reader expect responses to visual stimuli to be equivalent to responses to auditory stimuli?

3.       The authors should report the existence of a valid institutional protocol approval # for experiments with human subjects, and the informed consent mechanism that was used.

Some minor suggestions:

1.       This manuscript would benefit from professional editing for grammar and typos. Here are some suggestions but many more phrases could be improved for clarity.

2.       Replace “early intervention on infants that maximally take advantage” by “early intervention on infants that maximally takes advantage”.

3.       Replace “(FASD1;” by “(FASD1)”.

4.       Replace “toe stablish” by “to establish”.

5.       Replace “This result clearly show that” by “This result clearly shows that”

6.       Figure 4D,E shows the use of a car seat in conditions presumably deviating from safety standards and manufacturer instructions. Perhaps a note should be added to clarify this.

Reviewer 3 Report

This comminication presents: Wireless Heart Sensor for Capturing Cardiac Orienting Re- 2 sponse for Prediction of Neurodevelopmental Delay in Infants.

In my opinion it's interesting but I recommend that you should present the results more carefully. The description of the results should be clearer and refer to all the results obtained by the authors. In addition, the discussion lacks the presentation of own results against the background of the latest research. I also recommend that you include a short summary.

Reviewer 4 Report

Type of manuscript: Communication

Title: Wireless Heart Sensor for Capturing Cardiac Orienting Response for

Prediction of Neurodevelopmental Delay in Infants:

Reviewing: [Sensors] Manuscript ID: sensors-1832771

General Comment:

-         Well written Communication.  Short and concise. Interesting topic to be published. Still few comments to be looked into.

-          This is a short article. The length could be an issue, but since it is a communication type publication, this can be accepted. 

-         The Communication is not showing any of the sensed (heartbeat times) patterns. It will be much effective to show to collected ECG data.  How compared with standard ECG without wireless system. 

-          The Communication did not present any accuracy measurement and errors of the sensed dataset.  It will be much effective to present the error and accuracy of the developed sensory hardware.

-          I could not find any mathematical foundation for the sensory mechanism, … ? this could be an issue …  

-          Once again, accuracy of the developed hardware is to be well presented.

-          Figure 3. COR: it needs to be made clear.

-         Discussion Section:  needs to be made much focused, I could not find fundamental outcomes from the presented discussions.

-        References: Although this is a Commination Type publication, … but still I found that the references list is short. More about the hardware is be referenced.

Round 2

Reviewer 3 Report

Dear Authors,

now, the manuscript looks better. There are some litte mistakes in the text. You can read it carefully and correct them.